# The Use of Cabozantinib in Advanced Hepatocellular Carcinoma in Hong Kong—A Territory-Wide Cohort Study

**DOI:** 10.3390/cancers13092002

**Published:** 2021-04-21

**Authors:** Jeffrey Sum-Lung Wong, Yawen Dong, Vikki Tang, Thomas Leung, Cynthia S. Y. Yeung, Anna Tai, Ada Law, Tracy Shum, Gerry Gin-Wai Kwok, Bryan Cho-Wing Li, Roland Leung, Joanne Chiu, Ka-Wing Ma, Wong-Hoi She, Josephine Tsang, Tan-To Cheung, Thomas Yau

**Affiliations:** 1Department of Medicine, Queen Mary Hospital, The University of Hong Kong, Hong Kong, China; wsl714@ha.org.hk (J.S.-L.W.); yawen.dong@gesundheitsverbund.at (Y.D.); vyftang@hku.hk (V.T.); kgw951@ha.org.hk (G.G.-W.K.); lcw027@ha.org.hk (B.C.-W.L.); lcy035@ha.org.hk (R.L.); jwychiu@hku.hk (J.C.); twy917@ha.org.hk (J.T.); 2Department of Medicine, Hong Kong Sanatorium and Hospital, Hong Kong, China; thomas.wt.leung@hksh.com; 3Department of Clinical Oncology, Tuen Mun Hospital, Hong Kong, China; Syyeung@ha.org.hk; 4Department of Clinical Oncology, Queen Elizabeth Hospital, Hong Kong, China; typ065@ha.org.hk; 5Department of Clinical Oncology, Pamela Youde Nethersole Eastern Hospital, Hong Kong, China; lawly@ha.org.hk; 6Department of Clinical Oncology, Princess Margaret Hospital, Hong Kong, China; scy409@ha.org.hk; 7Department of Surgery, Queen Mary Hospital, The University of Hong Kong, Hong Kong, China; kawingma@hku.hk (K.W.-M.); swh180@ha.org.hk (W.H.-S.); cheung68@hku.hk (T.T.-C.)

**Keywords:** hepatocellular carcinoma, cabozantinib, tyrosine kinase inhibitors

## Abstract

**Simple Summary:**

The vascular endothelial growth factor and c-MET pathways are strongly implicated in hepatocellular carcinoma (HCC). Cabozantinib inhibits both pathways and has been approved in sorafenib-exposed advanced HCC (aHCC). We aimed to evaluate the real-life pattern of use, efficacy, and safety of cabozantinib in aHCC patients in a territory-wide study. In single-agent cabozantinib patients (*n* = 27), we found that 3.7% had a response, 44.4% had disease control, and the median overall survival (OS) was 8.28 months. Around 74.1% of patients had adverse events (AEs). We also did an exploratory analysis of patients who received cabozantinib as an add-on to immune-checkpoint inhibitors (ICIs) after progression on ICIs. Out of 15 such patients, 6.7% had a response and the median OS was 15.1 months, with 73.3% of patients having AEs. Overall, cabozantinib had good efficacy, survival, and safety in aHCC patients in a real-life setting.

**Abstract:**

(1) Background: Cabozantinib is approved in sorafenib-exposed advanced hepatocellular carcinoma (aHCC). We evaluated the real-life pattern of use, efficacy, and tolerability of cabozantinib in aHCC. (2) Methods: This territory-wide study included consecutive aHCC patients who received cabozantinib between February 2018 and September 2020 in Hong Kong. The objective response rate (ORR), disease control rate (DCR), overall survival (OS), and adverse events (AE) were assessed. (3) Results: Overall, 42 patients were included. Approximately 83.3% had Child-Pugh A cirrhosis. About 64.3% received cabozantinib as a single agent, and the remaining 35.7% received cabozantinib as an add-on to immune checkpoint inhibitors (ICIs). For single-agent patients, the median follow-up was 6.7 months. The ORR was 3.7%, DCR was 44.4%, and the median OS was 8.28 months. About 74.1% of patients experienced any AEs with 7.4% having grade ≥3 AEs. Among patients who received prior ICIs (*n* = 16), the ORR was 6.3%, and the median OS was 8.28 months. An exploratory analysis of patients who received cabozantinib as an add-on to ICIs showed an ORR of 6.7% and a median OS of 15.1 months, with 73.3% having any AE and 13.3% having grade ≥3 AEs. (4) Conclusions: Cabozantinib had good anti-tumor activity, survival benefits, and acceptable tolerability in real-life aHCC patients.

## 1. Introduction

The majority of hepatocellular carcinomas (HCCs) present as advanced disease not amenable to surgery [1]. Vascular endothelial growth factor (VEGF)-mediated cellular signaling pathways have been implicated in the pathogenesis of HCC and studied extensively as a therapeutic target [2]. Sorafenib, a small molecule tyrosine kinase inhibitor (TKI) that inhibits VEGF pathways, was the first agent licensed for use in the systemic treatment of HCC. In the landmark SHARP trial, sorafenib demonstrated a superior median overall survival (OS) (10.7 months vs. 7.9 months, *p* < 0.001) and median time to progression (5.5 months vs. 2.8 months, *p* < 0.001) compared to the placebo [3]. Lenvatinib demonstrated non-inferiority to sorafenib (median OS 13.6 months vs. 12.3 months) in the REFLECT trial and was duly licensed for use as first-line treatment of advanced HCC (aHCC) [4]. Regorafenib became the first ever agent to be licensed for use in second-line treatment for patients who progressed on sorafenib after demonstrating significant improvement in OS (hazard ratio 0.63, *p* < 0.0001, median OS 10.6 months vs. 7.8 months) compared to the placebo in this population in the RESORCE trial [5]. Finally, ramucirumab, a VEGFR-2 inhibitor, was licensed for use in sorafenib-treated aHCC with alpha-fetoprotein (AFP) ≥400 ng/mL after demonstrating superior OS (median OS 8.5 vs. 7.3 months, *p* = 0.0199) compared to the placebo in the REACH-2 trial [6].

The c-Met and AXL receptor tyrosine kinases promote epithelial-to-mesenchymal transition, invasion, and metastasis in human malignancies [7,8,9]. Additionally, the c-Met pathway has been found to be up-regulated in HCCs treated with sorafenib, implicating it in sorafenib resistance [10,11]. The multi-kinase inhibitor cabozantinib has activity against VEGF receptors 1-3, c-Met, and the TAM receptors (Tyro-3, AXL, and Mer), thus conferring it the theoretical benefit of overcoming sorafenib resistance in HCC [10,12,13]. In the CELESTIAL trial, cabozantinib demonstrated significantly superior overall survival (median OS 10.2 months vs. 8.0 months, *p* = 0.005), PFS (5.2 months vs. 1.9 months, *p* < 0.001), and objective response rate (4% vs. ≤1%, *p* = 0.009) compared to the placebo in patients with sorafenib-treated HCC [14]. This led to cabozantinib’s approval as a treatment for patients with sorafenib-treated HCC in second- or third-line settings.

Despite the encouraging results of the CELESTIAL trial, important questions regarding the use of cabozantinib in aHCC remain. Firstly, there may be significant differences in the outcome and safety of cabozantinib in real-life use compared to trial settings. These potential differences may be due to trial exclusion criteria such as clinical and laboratory limits on liver and hematological function [14] as well as the time needed for trial screening, both of which likely precluding a significant number of patients with advanced cirrhosis or aggressive HCC from trial participation. Secondly, since the official launch of cabozantinib worldwide, options of systemic HCC treatment have undergone a significant expansion. In particular, multiple immune checkpoint inhibitors (ICIs) have been approved both in first- and second-line settings. These include atezolizumab-bevacizumab in the first line based on the phase III IMbrave150 trial [15,16], nivolumab [17], nivolumab-ipilimumab [18], and pembrolizumab [19] as second-line treatment based on the single-arm phase I/II CheckMate-040 and Keynote-224 trials. However, it should be noted that nivolumab and pembrolizumab monotherapy did not reach pre-defined endpoints in the subsequent phase III studies [20,21]. Despite there being a small subgroup of CELESTIAL patients having received ICIs [22], the efficacy of cabozantinib after ICIs is largely unknown. We thus conducted a territory-wide study of aHCC patients who received cabozantinib in a real-life setting.

## 2. Methods

This was a multi-center, territory-wide retrospective analysis involving six tertiary/quaternary oncology centers in Hong Kong. The study was approved by the University of Hong Kong/Hospital Authority Hong Kong West Cluster Institutional Review Board and was conducted based on the principles of the Declaration of Helsinki. Between February 2018 and September 2020, consecutive patients with HCC not amenable to loco-regional therapy who received cabozantinib were included. Data were retrieved from the anonymized territory-wide electronic records of the Hong Kong Hospital Authority (HA), which covers 90% of all secondary and tertiary care in the territory.

HCCs were diagnosed either according to the European Association for the Study of the Liver (EASL) combined criteria when both an elevated serum AFP and characteristic radiological findings were present, or by histological confirmation with operative specimens, fine needle aspiration, or biopsy. Staging was done by computer tomography or magnetic resonance imaging according to the Barcelona Clinic Liver Cancer (BCLC) system. Relevant clinical, imaging, and laboratory information were gathered from the aforementioned source.

### 2.1. Dosing, Evaluation, and Follow-Up

The starting doses of cabozantinib in individual patients were decided by treating physicians. Tumor responses were evaluated according to both the Response Evaluation Criteria in Solid Tumor version 1.1 (RECIST 1.1) and the modified RECIST (mRECIST) for HCC [23,24]. The best objective response (BOR) was defined as the best response compared to baseline per RECIST 1.1 or mRECIST. The objective response rate (ORR) was defined as the percentage of patients with a complete response (CR) or a partial response (PR). The disease control rate (DCR) was defined as the percentage of patients with CR, PR, or stable disease (SD). Reassessment scans were generally scheduled every 8–12 weeks. For responses to prior ICIs, primary resistance was defined as always having progressive disease, while acquired resistance was defined as ever having stable or responding disease. Adverse events (AEs) were graded with the National Cancer Institute’s Common Terminology Criteria for Adverse Events (NCI CTCAE) version 4.0 [25]. Patients were followed up with regular physical examination, blood tests, and scanning.

### 2.2. Statistical Analysis

Statistical analyses were performed using SPSS (version 26, IBM, Armonk, NY, USA). Follow-up time was calculated from the date of the first dose to death or last follow-up. Overall survival (OS) was calculated from the date of the first dose to death from any cause or censored at the last follow-up if the patient was still alive. Time to progression (TTP) was calculated from the date of the first dose to tumor progression or censored at the last follow-up or death while on treatment for patients without progression. The median OS, TTP, and survival rates were estimated by Kaplan–Meier analyses, and survival curves were compared using log-rank tests. The median follow-up time was estimated using the reverse Kaplan–Meier method. Categorical variables were compared using Pearson’s chi-squared test or Fisher’s exact test. Nonparametric statistics were used to compare continuous variables. A *p*-value of 0.05 was taken as cut-off for statistical significance. 

## 3. Results

### 3.1. Demographics

Forty-two patients were included in the study. Table 1 and Appendix A show their baseline characteristics. The median age was 59.5 (range 41–85). Most patients had Child–Pugh (CP) grade A cirrhosis (83.3%), and 31% were of Albumin-Bilirubin (ALBI) grade 1. About 88.1% had hepatitis B-related HCC. Notably, 64.3% of analyzed patients received single-agent cabozantinib (*n* = 27), while the remaining 35.7% patients used cabozantinib as add-on therapy to ongoing ICIs (anti-PD1 with or without anti-CTLA-4) (Appendix A).

Regarding prior treatments, the patients in our cohort were generally heavily pre-treated. Around 35.7% of patients received cabozantinib as second-line therapy, and 21.4% of patients received cabozantinib as third-line therapy. The remaining 16.7% and 26.2% of patients received cabozantinib as their 4th and ≥5th line of systemic treatment, respectively. Of note, single-agent cabozantinib was most often used as second-line therapy, while cabozantinib–ICI combinations were more frequently used as salvage therapy in fourth line or beyond. The vast majority of patients were exposed to tyrosine kinase inhibitors (95.2%), with 66.7% exposed to lenvatinib, and 40.5% to sorafenib. Notably, 31 (73.8%) patients had prior exposure to ICIs before cabozantinib. The majority of patients started cabozantinib after progression on prior regimes, with some due to side effects. 

### 3.2. Outcomes of Single-Agent Cabozantinib

#### 3.2.1. Clinical Outcomes

The median follow-up was 6.74 months (95% C.I., 3.35–10.1). At the time of study-cut off, six patients were lost to follow-up. Table 2 summarizes the BORs. The ORR and DCR were 3.7% and 44.4%, respectively. There were no differences in the BORs by the RECIST 1.1 and mRECIST criteria. The median TTP was 3.88 months (95% C.I., 2.31–5.45). The median OS was 8.28 months (95% C.I., 3.91–12.7) (Figure 1). The 6-month and 1-year survival rates were 61.1% and 17.8%, respectively. 

#### 3.2.2. Dosing and Adverse Events

The median treatment duration was 2.27 months (range 0.2–11.7). The median daily dose was 40 mg (range 20–60). At the study cut-off, 44.4% of patients terminated treatment due to progressive disease, 14.8% due to intolerance, 3.7% due to non-cancer death, and 22.2% due to unrelated medical conditions and other causes. Around 14.8% of patients were still receiving treatment. About 25.9% of patients received other subsequent systemic treatments.

Table 3 shows all AEs that were observed in the study. The most common AEs were hand-foot syndrome (HFS), hypertension, malaise, diarrhea, and transaminitis. A total of 74.1% and 7.4% of patients had all grade and grade ≥3 AEs, respectively. One patient had grade 3 hypertension and another had grade 3 HFS. Overall, 22.2% of patients required dose interruptions, 25.9% required dose reduction, and 14.8% required treatment termination due to AEs. There were no treatment-related deaths.

#### 3.2.3. Single-Agent Cabozantinib Post-Immune Checkpoint Inhibitors

As mentioned above, 16 (59.3%) patients had prior exposure to ICIs before receiving single-agent cabozantinib. The median interval between the last dose of previous ICI and cabozantinib was 5.36 weeks (range 1.71–84). Three patients discontinued prior ICIs due to AEs, with the remaining discontinuations due to progressive disease. Approximately 37.5% of patients had acquired resistance, and 43.8% had primary resistance to prior ICI-containing regimes. Overall, the ORR was 6.3%, the DCR was 31.3%, the median TTP was 3.51 months (95% C.I., 1.22–5.82), and the median OS was 8.28 months (95% C.I., 1.33–15.2) (Table 4). There was no significant difference in the median OS among patients with primary resistance to prior ICI regimes compared to those with acquired resistance (primary resistance 3.52 months (95% C.I., 0–10.0), acquired resistance 7.03 months (95% C.I., 0–15.9), *p* = 0.891).

### 3.3. Exploratory Analysis of Patients on Cabozantinib–ICI Combinations

In real-life practice, 15 patients (35.7%) received cabozantinib in combination with ICIs. All of them had cabozantinib added to the ongoing ICI regime as add-on therapy after having progressive disease on the prior ICI regime. Add-on patients received prior ICIs for a longer time (the median duration on previous ICIs was 44.1 weeks (range 8–184.6), vs. 15.25 weeks (range 2–46), *p* = 0.009) compared to single-agent post-ICI patients. In this population, an overall ORR of 6.7% and DCR of 26.7% were observed (Table 4). The median TTP was 2.27 months (95% C.I., 1.40–3.14) and median OS was 15.1 months (95% C.I., 11.1–19.2) (Figure 1). The 6-month and 1-year survival rates were 79.4% and 71.5%, respectively. A total of 73.3% of such patients had any grade AEs while 13.3% had grade ≥3 AEs. The most common AE was HFS. Twenty percent of patients required a dose reduction and 13.3% required treatment cessation due to AEs (Table 3). 

## 4. Discussion

In our present study, we reported an ORR of 3.7%, DCR of 44.4%, median TTP of 3.88 months, and a median OS of 8.28 months in aHCC patients who received single-agent cabozantinib in a real-life setting. Importantly, our cohort provided salient data on the use of cabozantinib in a contemporary set of patients who progressed after various recently approved TKIs and ICIs. We also observed an interesting pattern of real-life use of cabozantinib as an add-on therapy to patients who progressed on ICIs.

The VEGF/VEGFR pathway is heavily implicated in HCC pathogenesis [2]. VEGFR-2 binds to VEGF-A to D, causing endothelial proliferation and angiogenesis and facilitating tumor progression [2]. Tumors with high VEGF levels have higher micro vessel density, more aggressive progression, and poorer survival [26,27,28]. Blockade of VEGF pathways can lead to up-regulation of the c-Met pathway. c-Met is a tyrosine kinase receptor for hepatocyte-growth factor (HGF) [29]. It is induced by hypoxia-inducible factor-1 (HIF-1) and involved in hepatic repair in chronic liver disease by inducing hepatocyte regeneration and suppressing chronic inflammation [30,31,32,33]. However, aberrant c-Met activation drives HCC pathogenesis and progression through promotion of cellular proliferation, invasion, and resistance to VEGF pathway blockade [9,34]. Cirrhosis increases c-MET activity through inducing HIF-1 by tissue hypoxia and increasing hepatocyte regeneration [29]. HCC further increases c-Met activity through increasing genomic instability and HGF levels [29,35]. In fact, c-Met aberrations are found in ~50% of all HCCs [29]. By primarily targeting VEGFR-2 and c-Met pathways, cabozantinib can block multiple oncogenic, angiogenic, and escape pathways and exert anti-tumor effects on HCCs [10]. 

In the CELESTIAL trial, an ORR of 4% and a median OS of 10.2 months were observed with single-agent cabozantinib [14]. In another single-arm phase II trial, Kudo et al. reported an ORR of 0%, DCR of 76.5%, and an OS rate at 6 months of 91.1% in 34 patients with aHCC who were mainly pretreated with sorafenib or lenvatinib, and with cabozantinib exclusively used in the second- or third-line setting [36]. Two other studies also reported the real-life use of single-agent cabozantinib [37,38]. In a cohort of 52 patients with CP-A liver function, mostly (80.8%) aHCC, and with cabozantinib mostly in a third-line setting (88.0%), Tovoli et al. reported a DCR of 59.2% and a median OS of 12.9 months. Meanwhile, Finkelmeier et al. reported a median OS of 7.7 months in a cohort of 74 patients with cabozantinib mainly as second- (50%) and third-line (34%) treatment, with 5% and 30% of patients achieving partial response and stable disease, respectively. Generally speaking, the results from our analysis were quite comparable with similar response rate and survival. Furthermore, it is worthwhile to note that single-agent cabozantinib was, in general, well tolerated in our study, with grade 3–4 AEs occurring in only 7.4% of such patients. The profile of AEs was similar to other published studies of cabozantinib. Interestingly, the rates of grade 3–4 AEs and AEs requiring dose reduction were all lower in our study than in other published trials [14,36]. This may reflect the differences in collection of AEs in real-life practice compared to clinical trial settings. An alternative explanation is that this may be due to a difference in dosing strategies in a real-life setting compared to in clinical trials, as cabozantinib was generally started at lower dose and titrated up to 60 mg daily if tolerated in our patients, whilst treatment was generally commenced at 60 mg directly in trials with subsequent dose reduction if needed. In fact, the majority of our patients only received an average daily dose of 40 mg or below.

There is an urgent need to establish effective therapy for patients who are refractory to ICI based treatments. In the CELESTIAL trial, 14 patients who had received prior sorafenib and anti-PD-1/L1 were put on cabozantinib. In this subgroup, a median OS of 7.9 months and 64% of patients having grade 3 or 4 AEs were reported [22]. Post-ICI TKIs were also recently described in patients who received sorafenib, lenvatinib, or cabozantinib as second-line treatment after first-line atezolizumab-bevacizumab, demonstrating an ORR of 6.1%, DCR of 63.3%, and a median OS of 14.7 months. However, only one patient with cabozantinib was included in this study [39]. To our knowledge, no other literature exists in regard to post-ICI use of cabozantinib. However, several phase II trials for single-agent cabozantinib in this setting are currently underway [40,41]. In our study, we demonstrated that post-ICI use of single-agent cabozantinib can achieve good tumor control, survival, and tolerability. Moreover, in real-life practice, cabozantinib was also used as add-on therapy to ICIs. Notably, those who received cabozantinib as add-on therapy had been treated with prior ICI for a longer duration. This may be due to clinicians seeing benefits in continuing ICIs beyond progression due to good tolerability and indolent disease progression. Therefore, they were more inclined to add cabozantinib to ICIs rather than switching to other new therapies even after progression. In fact, there is an increasing amount of evidence of synergism between cabozantinib and ICIs in the literature. It was recently shown that c-Met inhibition by tivantinib/capmatinib reduces GSK3B-mediated PDL-1 degradation, resulting in increased PDL-1 expression and inactivation of co-cultured T cells in the HCC cell line, as well as reduced anti-tumor activity of T cells in mouse models [42]. Compared to anti-PD1 or c-Met inhibitors alone, combined treatment of c-Met inhibitors with anti-PD-1 increased tumor-infiltrating CD8+ T cells, decreased tumor growth, and prolonged survival of such mice [42]. Moreover, early clinical trial results demonstrated synergistic action of cabozantinib plus ICIs in aHCC. In the results of the CheckMate-040 trial cohort 6, nivolumab-cabozantinib and nivolumab-ipilimumab-cabozantinib demonstrated ORRs of 17% and 26%, DCRs of 81% and 83%, as well as a median PFS of 5.5 and 6.8 months, respectively [43]. The median OS was 21.5 months for the doublet arm and not reached for the triplet arm, respectively. Additionally, the COSMIC-312 trial, which assessed the efficacy and safety of atezolizumab/cabozantinib as first-line treatment in aHCC patients, completed its recruitment [44]. The mature results from COSMIC-312 will provide the HCC community with important information about potential synergistic activities of cabozantinib with ICIs. Although the involved number of patients were small, the efficacy, survival, and tolerability achieved by this approach was encouraging. Nevertheless, it should be noted that the practice and benefits of using add-on cabozantinib to ICIs is not supported by any available phase III clinical trial data and was only an interesting phenomenon that we observed in this real-life study.

This study has several limitations, including small sample size, heterogenous population, biases inherent in its retrospective nature, and lack of blinded, independent review of treatment responses. The subgroup analyses contained a small number of patients, and large-scale studies are needed to validate their findings. Owing to limitations inherent to real-life practice, reassessment imaging was only planned every 8–12 weeks instead of every 6 weeks per available data. Additionally, the majority of our patients only received 20–40 mg/day of cabozantinib instead of the registration dose of 60 mg/day due to safety concerns in a real-life setting. Furthermore, the usage of cabozantinib as a single agent beyond the third line, or with ICIs beyond the first line, is not evidence-based and needs further clarification. The cabozantinib combinations used were heterogenous and have not been validated beyond the preclinical stage. These factors limit the study to hypothesis-generating only and precludes direct recommendation of the use of cabozantinib in aHCC beyond its current licensed indication.

## 5. Conclusions

In conclusion, this study demonstrated that real-life use of cabozantinib can achieve encouraging anti-tumor activity and survival outcomes with good tolerability in aHCC patients in second line or beyond. Importantly, we also showed that post-ICI use of cabozantinib either as a single agent or as add-on therapy to ICIs can achieve tumor control, good safety profiles, and potential survival benefits. The results of ongoing prospective studies will verify our findings and address the unmet medical needs.

## Figures and Tables

**Figure 1 cancers-13-02002-f001:**
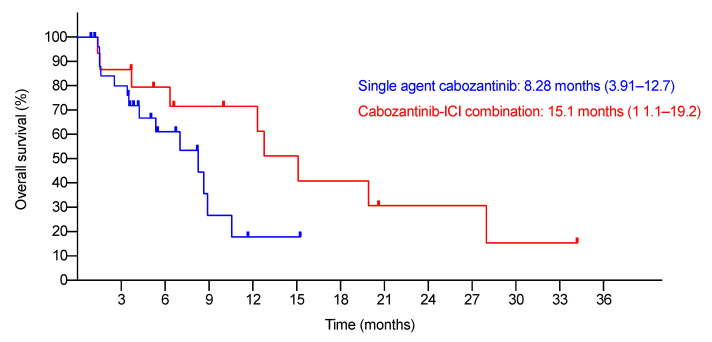
Kaplan–Meier analysis of OS of single-agent and cabozantinib–ICI combination patients.

**Table 1 cancers-13-02002-t001:** Baseline characteristics.

		All Patients (*n* = 42)	Single Agent (*n* = 27)	Combination (*n* = 15)
Median age (range), years		59.5 (41–85)	58 (41–85)	64 (42–79)
Male, *n* (%)		36 (85.7%)	25 (92.6%)	11 (73.3%)
Hepatocellular carcinoma etiology, *n* (%)	Hepatitis B (HBV)	37 (88.1%)	23 (85.2%)	14 (93.3%)
	Hepatitis C (HCV)	2 (4.8%)	1 (3.7%)	1 (6.7%)
	Alcoholic	1 (2.4%)	1 (3.7%)	0
	Non-alcoholic Steatohepatitis (NASH)	2 (4.8%)	2 (7.4%)	0
Child-Pugh, *n* (%)	A	35 (83.3%)	23 (85.2%)	12 (80%)
	B	6 (14.3%)	4 (14.8)	2 (13.3%)
	C	1 (2.4%)	0	1 (6.7%)
Albumin-Bilirubin Grade, *n* (%)	1	13 (31%)	7 (25.9%)	6 (40%)
	2	24 (57.1%)	16 (59.3%)	8 (53.3%)
	3	5 (11.9%)	4 (14.8%)	1 (6.7%)
Barcelona Clinic Liver Cancer stage, *n* (%)	B	2 (4.8%)	1 (3.7%)	1 (6.7%)
	C	39 (92.9%)	26 (96.3%)	13 (86.7%)
	D	1 (2.4%)	0	1 (6.7%)
Baseline Performance Status, *n* (%)	01	40 (95.2%)	27 (100%)	13 (86.7%)
	2	2 (4.8%)	0	2 (13.3%)
Alpha-fetoprotein ≥400 ng/mL, *n* (%)		21 (50%)	13 (48.1%)	8 (53.3%)
Extrahepatic metastases, *n* (%)		37 (88.1%)	24 (88.9%)	13 (86.7%)
Vascular invasion, *n* (%)		7 (16.7%)	6 (22.2%)	1 (6.7%)
Cabozantinib setting, *n* (%)	Second line	15 (35.7%)	13 (48.1%)	2 (13.3%)
	Third line	9 (21.4%)	6 (22.2%)	3 (20%)
	Fourth line	7 (16.7%)	3 (11.1%)	4 (26.7%)
	Fifth line and beyond	11 (26.2%)	5 (18.5%)	6 (40%)
Previous therapies, *n* (%)	Curative resection	21 (50%)	9 (33.3%)	12 (80%)
	Radiotherapy	10 (23.8%)	6 (22.2%)	4 (26.7%)
	Transarterial chemoembolization	20 (47.6%)	13 (48.1%)	7 (46.7%)
	Any tyrosine kinase inhibitors	40 (95.2%)	26 (96.3%)	14 (93.3%)
	Any immune checkpoint inhibitors	31 (73.8%)	16 (59.3%)	15 (100%)

**Table 2 cancers-13-02002-t002:** Best objective response, all patients.

Activity, *n* (%)	All Patients (*n* = 42)	Single Agent (*n* = 27)	Combination (*n* = 15)
Progressive disease	18 (42.9%)	10 (37.0%)	8 (53.3%)
Stable disease	14 (33.3%)	11 (40.7%)	3 (20%)
Partial response	2 (4.8%)	1 (3.7%)	1 (6.7%)
Complete response	0 (0%)	0	0
Non-evaluable	8 (19.0%)	5 (18.5%)	3 (20%)
Objective response rate	4.8%	3.7%	6.7%
Disease control rate	38.1%	44.4%	26.7%

**Table 3 cancers-13-02002-t003:** Adverse events (AEs).

		All Patients (*n* = 42)	Single Agent (*n* = 27)	Combination (*n* = 15)
All AE		31 (73.8%)	20 (74.1%)	11 (73.3%)
≥Grade 3 AE		4 (9.5%)	2 (7.4%)	2 (13.3%)
AE requiring dose interruption		8 (19.0%)	6 (22.2%)	2 (13.3%)
AE requiring dose reduction		10 (23.8%)	7 (25.9%)	3 (20%)
AE requiring treatment cessation		6 (14.3%)	4 (14.8%)	2 (13.3%)
Hand-Foot Syndrome				
	Grade 1-2	12 (28.6%)	8 (29.6%)	4 (26.7%)
	Grade 3	2 (4.8%)	1 (3.7%)	1 (6.7%)
Hypertension				
	Grade 1-2	6 (14.3%)	5 (18.5%)	1 (6.7%)
	Grade 3	2 (4.8%)	1 (3.7%)	1 (6.7%)
Transaminitis		6 (14.3%)	4 (14.8%)	2 (13.3%)
Diarrhea		6 (14.3%)	5 (18.5%)	1 (6.7%)
Nausea/vomiting		3 (7.1%)	2 (7.4%)	1 (6.7%)
Malaise/Loss of appetite		7 (16.7%)	6 (22.2%)	1 (6.7%)
Rash/Other Skin issues		6 (14.3%)	4 (14.8%)	2 (13.3%)
Musculoskeletal pain		2 (4.8%)	1 (3.7%)	1 (6.7%)
Proteinuria		5 (11.9%)	4 (14.8%)	1 (6.7%)
Neutropenia		4 (9.5%)	4 (14.8%)	0
Thrombocytopenia		3 (7.1%)	3 (11.1%)	0
Epigastric Discomfort		1 (2.4%)	1 (3.7%)	0
Palpitations		1 (2.4%)	1 (3.7%)	0
Hypothyroidism		2 (4.8%)	1 (3.7%)	1 (6.7%)

**Table 4 cancers-13-02002-t004:** Best objective response, post-immune checkpoint inhibitors patients.

Activity, *n* (%)	All Patients (*n* = 31)	Single Agent (*n* = 16)	Combination (*n* = 15)
Progressive Disease	15 (48.4%)	7 (43.8%)	8 (53.3%)
Stable Disease	7 (22.6%)	4 (25%)	3 (20%)
Partial Response	2 (6.5%)	1 (6.3%)	1 (6.7%)
Complete Response	0	0	0
Non-evaluable	7 (22.6%)	4 (25%)	3 (20%)
Objective Response Rate	6.5%	6.3%	6.7%
Disease Control Rate	29.0%	31.3%	26.7%

## Data Availability

The data presented in this study are not publicly available for privacy and legal reasons.

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
