# Peer review of "The Use of Cabozantinib in Advanced Hepatocellular Carcinoma in Hong Kong—A Territory-Wide Cohort Study"

_cancers, 2021, doi:10.3390/cancers13092002_

Round 1

Reviewer 1 Report

This a real life study well performed confirming data of the trials.

The study proposes a real life study on a drug whose clinical trial showed being efficient and well tolerated in advanced sorafenib exposed HCC patients.

The study deserves publication because it responds to the need of well organized real life study to confirm data from randomized trials. Moreover, authors describe with accuracy and details either the introduction and the discussion supporting their results. This study offers another piece of evidence among all the other anti neoplastic agents for advanced HCC as second line treatments whose need is in continuous increment trying to find the best clinical equilibrium between efficacy and tolerability. 

Reviewer 2 Report

Cancers 1175034

Wong JSL. et al. analyzed the impact of the cabozantinib for the advanced HCC in Hong Kong. This manuscript has some messages to the readers of the cancers, but some refinements are needed for this manuscript.

1)    In the methods section, the description of the Declaration of Helsinki and IRB are necessary.

2)    In the results section, the authors should describe the precise reasons of the changing agents before administrating cabozantinib.

3)    In the fig 1., the authors should show the results of the combination patients.

Reviewer 3 Report

They showed that the cabozantinib can achieve encouraging anti-tumour activity and survival outcomes with good tolerability in HCC patients. Their sample size was a relatively small, but this finding was quite important for severe HCC patients who didn’t have any other treatment.

I have some comments.

  1. How did you diagnose the relationship between their adverse events and cabozantinib?
  2. How did you decide single agent or combination therapy? What was this indication?

Round 2

Reviewer 2 Report

Cancers 1175034 R1

The revised manuscript was corrected in accordance with the suggestions of the reviewers.